# Adding pattern and process to eco-evo theory and applications

**Jennifer M. White[1], Nathan H. Schumaker[2]\*, Rachel Y. Chock[3], Sydney M. Watkins[4]**

**1** Center for Conservation Biology, Department of Biology, University of Washington, Seattle, Washington, United States of America, **2** U.S. Environmental Protection Agency, Pacific Ecological Systems Division, Corvallis, Oregon, United States of America, **3** San Diego Zoo Wildlife Alliance, Conservation Science, Escondido, California, United States of America, **4** Oak Ridge Institute for Science and Education, Oak Ridge, Tennessee, United States of America

\* nathan.schumaker@gmail.com

**Data Availability Statement:** Our study is heuristic, and does not make use of any actual empirical data. However, our entire model, including all requisite inputs, are freely available at www.hexsim.net: Schumaker, Nathan H. (2023).

## Abstract

Eco-evolutionary dynamics result when interacting biological forces simultaneously produce demographic and genetic population responses. Eco-evolutionary simulators traditionally manage complexity by minimizing the influence of spatial pattern on process. However, such simplifications can limit their utility in real-world applications. We present a novel simulation modeling approach for investigating eco-evolutionary dynamics, centered on the driving role of landscape pattern. Our spatially-explicit, individual-based mechanistic simulation approach overcomes existing methodological challenges, generates new insights, and paves the way for future investigations in four focal disciplines: Landscape Genetics, Population Genetics, Conservation Biology, and Evolutionary Ecology. We developed a simple individual-based model to illustrate how spatial structure drives eco-evo dynamics. By making minor changes to our landscape's structure, we simulated continuous, isolated, and semi-connected landscapes, and simultaneously tested several classical assumptions of the focal disciplines. Our results exhibit expected patterns of isolation, drift, and extinction. By imposing landscape change on otherwise functionally-static eco-evolutionary models, we altered key emergent properties such as gene-flow and adaptive selection. We observed demo-genetic responses to these landscape manipulations, including changes in population size, probability of extinction, and allele frequencies. Our model also demonstrated how demo-genetic traits, including generation time and migration rate, can arise from a mechanistic model, rather than being specified *a priori*. We identify simplifying assumptions common to four focal disciplines, and illustrate how new insights might be developed in eco-evolutionary theory and applications by better linking biological processes to landscape patterns that we know influence them, but that have understandably been left out of many past modeling studies.

## Introduction

Eco-evolutionary dynamics are shaped by spatial patterns and mediated by movement. The influence of spatial pattern on eco-evolutionary (hereafter "eco-evo") dynamics spans spatio-

HexSim Genetics Workspace. Zenodo. https://doi.org/10.5281/zenodo.7662662.

**Funding:** The authors received no specific funding for this work.

temporal scales and scientific disciplines, from cancer development within populations of somatic cells [1] to ecosystem responses to climate change (e.g., [2, 3]). With advances in molecular genetics, the time scales between evolutionary and ecological forces have been shown to be sufficiently coincident [4] and feedbacks from rapid-evolution impinging on ecological processes have been directly observed in an increasing number of natural systems [5–11]. For example, evolutionary geneticists follow the evolution of heritable traits responding to spatially heterogeneous selection pressures, population geneticists examine changes in allele frequencies influenced by spatially-restricted gene-flow, phylogeographers investigate coalescent timing across glaciations, and conservation biologists explore links between genetic diversity and resilience in spatially isolated populations. Despite the inherently spatial nature of these processes, the dynamics of landscape structure have proven difficult to incorporate into predictive models; though, for some exceptions see [12, 13]. Nevertheless, landscape dynamics may be critical for understanding reciprocal feedbacks between ecology and evolution [14], and for informing management applications [15].

Here, we illustrate how spatial structure drives eco-evo dynamics within four focal disciplines: Landscape Genetics, Population Genetics, Conservation Biology, and Evolutionary Ecology. We employ computer simulations to illustrate these dynamics using a novel modeling environment called HexSim [16] in which both biological forces and observable demographic and genetic responses emerge mechanistically from changes to landscape structure. HexSim life history processes are directly linked to static or dynamic landscape maps, and multiple spatial drivers can simultaneously influence different parts of the same simulation. Further, movement responses to landscape structure are not constrained by a reliance on resistance surfaces, patch-mosaic structures, stepping stones, or the use of graph-theoretic networks–all simplifications that are frequently employed to speed model development, but at a cost to biological realism. Lastly, HexSim eco-evo processes are mechanistically connected using a highly flexible system of demographic and genetic (hereafter "demo-genetic") life history traits. While more than 60 publications have resulted from HexSim-based studies (www.hexsim.net), adoption and application of the model's genetics toolkit has lagged. Below we describe how increasing attention to spatial and demographic details, and the further integration of ecological and evolutionary processes, will likely contribute to the focal disciplines mentioned above. Both the HexSim application and the simulation models described here are available at www.hexsim.net.

## Landscape genetics

This relatively new field endeavors to describe how landscape pattern influences gene-flow. A standard approach is to compare inter-individual genetic distance to metrics of landscape structure, often quantified as a cost-distance across a resistance surface. Available landscape genetics modeling platforms have had limited spatial, demographic, or behavioral sophistication [17], though some such constraints are being actively addressed by leaders in the field (e.g., [18]). Models that are unable to incorporate mechanisms underlying species-landscape interactions have a limited ability to simulate complex movements and resulting gene-flow. Additionally, most platforms cannot simultaneously simulate multiple interacting eco-evo drivers of gene-flow, such as local adaptation along with source-sink demographics [19].

## Population genetics

Investigators study the causes and consequences of population genetic structure. For simplicity, complex biological forces such as selection and mutation rates are often assumed to be spatially constant. Additionally, in classic population genetic models, spatial complexity is

reduced to discrete populations, and migration processes are characterized by a single univariate input parameter, '*m*'. These simplifications have facilitated the generation of a wealth of theory; but, a lack of biological realism and spatial complexity have precluded the application of this theory to systems in which space is heterogeneous and dynamic (e.g., where landscape structure influences population size, density, dispersal flux, or local adaptation), or when gene-flow is governed by complex movement and mating behavior (e.g., when dispersal rates and distances are unequal, or mate-choice is non-random; see [20] for review). However, exciting methodological advances are beginning to make estimates of population genetics structure more spatially realistic (e.g., [21]).

## Conservation biology

This broadly defined discipline frequently involves the exploration of forces affecting population viability. Landscape pattern drives population viability through its influence on ecological and evolutionary processes. The relative importance of demography versus genetics has been actively debated in the conservation literature [22, 23]. Some forecasting tools can simulate either inbreeding rates or landscape use, and incorporate them into probabilities of extinction (e.g., *RAMAS GIS* [24] or *Vortex* [25]), but existing models simplify the linkages between ecological and evolutionary processes, and these shortcomings limit the utility of their forecasts. For example, a platform capable of simulating complex interacting demographic and genetic traits is necessary to forecast how landscape changes will alter a population's size and distribution (eco), while simultaneously impacting its mate-finding or natural selection (evo). Sophisticated analyses of this sort have been conducted using traditional software applications, for example see [26, 27], but methodological advancements such as those described here are beginning to simplify these types of investigations.

## Evolutionary ecology

Researchers explore interactions and feedback between ecological and evolutionary drivers that affect demographic and genetic traits. Landscape patterns shape myriad mechanisms through which biological forces influence population and community demo-genetic traits. Renewed interest in these processes (e.g., [28, 29]) has underscored the need for biologically sophisticated, mechanistic simulation platforms capable of explicitly modeling dynamic eco-evo feedback in a spatially-realistic setting. Simply modeling how environmental change impacts heritable traits is only half of the story; our long-term goals must also include asking how these emergent genetic traits connect back to ecological dynamics [30, 31]. Recognition of these concerns has led to the development of several powerful new methods and software applications [32–35].

Below, we describe a relatively simple theoretical model designed to illustrate how the explicit inclusion of detailed spatial patterns and biologically realistic processes might facilitate the development of new eco-evo theory and applications. We demonstrate model relevance by highlighting results that (a) illustrate *a priori* expectations of a core eco-evo dynamic fundamental to a discipline, and (b) illustrate how additional biological and spatial realism might contribute to new theory and improve our confidence in modeling applications (Table 1). Our primary goals include both evaluating and articulating the benefits derived from adding spatial structure to genetic models, and illustrating the mechanics involved in doing this well. While the incorporation of space can substantially enhance realism and defensibility, it also complicates models, especially those which are already mechanistically-rich (e.g., that integrate demography, movement, and gene flow). In pursuit of the former goal, we trade away elements of realism, in the form of landscape structure and movement behavior, for mechanistic clarity.

**Table 1. How spatially structured eco-evolutionary models might contribute to four focal disciplines.**

| Discipline | Research Questions | Expected Outcomes | Anticipated Contributions |
|---|---|---|---|
| **Landscape Genetics** | How is gene-flow controlled by the landscape? | Isolation by Distance is a function of dispersal behavior. | Replacing resistance surfaces and cost-distance matrices with measures of genetic distance arising from eco-evo processes and species-landscape interactions. |
| **Population Genetics** | How is genetic structure controlled by the landscape? | Genetic structure is influenced by isolation by distance and demography. | Allowing migration rates between populations to emerge mechanistically from the interplay between dispersal behavior and landscape structure. |
| **Conservation Biology** | How are inbreeding and population viability controlled by the landscape? | Homozygosity and stochastic extinction are consequences of demographic processes. | Ensuring that forecasts of genetic degradation are driven by spatially-realistic movement models and incorporate pre-existing genetic structure and diversity. |
| **Evolutionary Ecology** | How are feedbacks between ecological and heritable traits controlled by the landscape? | Population demo-genetic traits are an ecological response to the evolutionary force of selection. | Creating eco-evo feedback loops between local selection for specific alleles and counteracting asymmetric migration stemming from source-sink dynamics in heterogeneous landscapes. |

Readers evaluating our assertions will benefit from the simplicity of our simulations, as the number of drivers and responses they must track is limited. In regards to the latter goal, our work constitutes an important extension to the existing descriptions and applications of Hex-Sim [16]. Readers interested in adopting this software will benefit from the case studies discussed here, as they provide detail on HexSim genetics that is unavailable in previous publications.

## Methods

Our study was built around a simple heuristic simulation constructed within the HexSim modeling environment [16]. HexSim is a user-friendly, spatially-explicit, individual-based, demo-genetic modeling environment. We modeled a single species, and parameterized its life history, demographics, genetic traits, and the interactions between habitat type and genetics (described below). We created a minimally-complex landscape structure that varied episodically throughout our simulations. Each epoch (1,000 simulation time steps) included unique barriers, barrier gaps, or habitat types. We varied behavior (dispersal distance), strength of selection, and barrier gap permeability for a total of eight treatment combinations, and tracked individual genotypes, per-capita homozygosity, per-patch population size, and the number of dispersers moving between the patches in each treatment. Through the use of this single model, we were able to probe expected outcomes of the four focal disciplines, and test the anticipated behaviors of this eco-evo modeling approach.

### Landscape structure

Our landscape was composed of six adjacent habitat patches (two small, two intermediate, two large) built up from multiple hexagonal cells of uniform quality (Fig 1). Two large adjoining patches (1326 hexagons each) lie at the landscape center. Two medium-sized patches (200 hexagons each) and two small patches (50 hexagons each) abut the large patches, but not each other. Patch dimensions expressed as columns × rows, were 26 × 51, 10 × 20, and 5 × 10. Collectively, the six patches were assembled from 3152 individual hexagons. Our simulated individuals were never allowed to enter the surrounding non-habitat matrix. Movement barriers were used, at times, to isolate the patches from each other. At other times, small semi-permeable openings in the barriers allowed limited dispersal between neighboring patches. The total size of the barrier gaps between adjacent patches was identical, ensuring individuals had a uniform potential for crossing all patch interfaces.

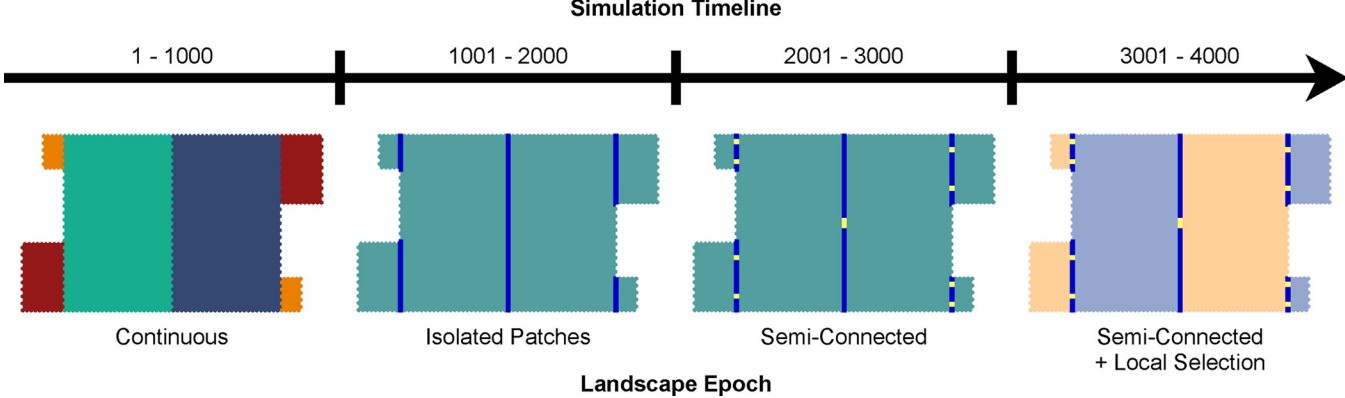

**Fig 1. The progression of landscape epochs over 4000 simulation time steps.** Movement barriers (shown as dark blue lines) and landscape edges are reflective. The six patches are distinguished by color within the *Continuous* epoch. Yellow barrier gaps, introduced at time step 2001, are of the same total length for each barrier. At time step 3001, the landscape is assigned two distinct habitat types (light blue vs. orange) that confer genotype-specific adaptive advantage to juvenile survival.

### Individual ecological characteristics

Our simulations ran for a series of time steps, each corresponding to one year, with a sequence of life history events and species-landscape interactions performed at each step. Note that we subsequently use the terms "time step" and "year" synonymously. We favor "time step" when describing model mechanics, and "year" when interpreting simulation output. In brief, the life cycle consisted of (a) resource acquisition, (b) pair formation, (c) reproduction, (d) juvenile dispersal, and (e) survival. The individuals making up our population included both sexes and two age classes (juvenile and adult), and our simulations began with the landscape being saturated with adults (3152 individuals spread uniformly across all patch hexagons). Adult females reproduced only if they could pair exclusively with an adult male located nearby (neither females or males had multiple mates). Individuals were assigned normally distributed reproductive rates, with mean values based on their resource allocation, which introduced a density-dependent feedback that was further modified by the requirement for pair-formation. Individuals acquired resources from a roughly-circular neighborhood of 37 hexagonal cells, and resources were shared equally by all individuals attempting to utilize them (scramble competition). Juveniles dispersed from their natal site in the year of their birth, and transitioned to adults at the start of the subsequent year. Adults did not move. Dispersal path lengths were drawn from one of two uniform distributions, either "short" (1–5 hexagon steps) or "long" (5–25 hexagon steps). Dispersal autocorrelation was set at 75, on a scale of 0 (completely random) to 100 (perfectly linear). Individual dispersers took a series of steps from hexagon to adjacent hexagon, and stopped when their path length had been reached. Yearly survival probability was based on stage class (juvenile = 0.500, adult = 0.885), but individuals with less than 20% of their resource goal were assigned an additional 10% probability of mortality. Our simulations included a period during which survival decisions were also based on genetic adaptation. In these cases, emergent mortality rates became jointly determined by stage class, location, and genotype (see below).

### Individual evolutionary characteristics

Our simulated individuals were diploid with ten loci and zero linkage between loci. Each locus was assigned five alleles labeled A1-A5. The starting population's allele assignment was governed by locus-specific initial allele frequencies (Table 2). Initial allele frequencies were not

**Table 2. Initial allele frequencies for 10-locus genotypes.**

| Locus | Allele | Initial Allele Frequency | Local Adaptation |
|---|---|---|---|
| 1 | 1–5 | 0.20, 0.20, 0.20, 0.20, 0.20 | Neutral |
| 2 | 1–5 | 0.30, 0.25, 0.20, 0.15, 0.10 | Neutral |
| 3 | 1–5 | 0.10, 0.15, 0.20, 0.25, 0.30 | Neutral |
| 4 | 1–5 | 0.01, 0.04, 0.15, 0.30, 0.50 | Neutral |
| 5 | 1–5 | 0.50, 0.30, 0.15, 0.04, 0.01 | Neutral |
| 6 | 1 | 0.20 | Neutral |
|   | 2 | 0.20 | Locally Adapted—Habitat Type A |
|   | 3 | 0.20 | Neutral |
|   | 4 | 0.20 | Locally Adapted—Habitat Type B |
|   | 5 | 0.20 | Neutral |
| 7 | 1 | 0.30 | Neutral |
|   | 2 | 0.25 | Neutral |
|   | 3 | 0.20 | Neutral |
|   | 4 | 0.15 | Neutral |
|   | 5 | 0.10 | Locally Adapted—Habitat Type A |
| 8 | 1 | 0.10 | Neutral |
|   | 2 | 0.15 | Neutral |
|   | 3 | 0.20 | Neutral |
|   | 4 | 0.25 | Neutral |
|   | 5 | 0.30 | Locally Adapted—Habitat Type B |
| 9 | 1 | 0.01 | Locally Adapted—Habitat Type A |
|   | 2 | 0.04 | Neutral |
|   | 3 | 0.15 | Neutral |
|   | 4 | 0.30 | Neutral |
|   | 5 | 0.50 | Neutral |
| 10 | 1 | 0.50 | Locally Adapted—Habitat Type B |
|   | 2 | 0.30 | Neutral |
|   | 3 | 0.15 | Neutral |
|   | 4 | 0.04 | Neutral |
|   | 5 | 0.01 | Neutral |

spatially stratified, and we did not simulate mutation. Offspring genotypes were assembled by drawing a single allele from each parent at each locus. Individuals possessed five purely neutral loci (L1-L5), and five loci (L6-L10) containing at least one allele capable of conferring a fitness advantage. These locally adaptive alleles imparted a survival benefit to juveniles when they were in the habitat type to which they were genetically pre-adapted. The survival benefit ($S$) per adaptive allele was either strong ($S = 0.10$) or weak ($S = 0.01$). Modeled this way, selection was a predictable process because genotype dictated the mean juvenile survival probability. Selection was initiated at time step 3000. All ten loci were utilized for all genetic analyses described below, with the exception of adaptation, for which only L6-L10 were employed.

## Progression of landscape change

All of our landscapes were binary, consisting of habitat patches embedded within a non-habitat matrix. Our initial landscape was continuous and free from movement barriers. This landscape, hereafter referred to as *Continuous*, persisted for the first epoch of 1000 simulation time steps (Fig 1). We anticipated that Isolation by Distance (IBD) would be the predominant

evolutionary force in this landscape. To simulate the effect of drift alone, we then imposed absolute movement barriers that isolated subpopulations into six separate landscape patches (two each, of three different sizes) for the subsequent 1000 simulation time steps. We refer to this epoch as *Isolated Patches*.

Following patch isolation, we created small gaps in the movement barriers, allowing infrequent migration between the previously isolated sub-populations. Barrier gaps varied in permeability (high = 0.70 transmission probability per encounter, low = 0.02 transmission probability per encounter), affecting the likelihood that individuals would cross the barrier when they encountered a gap during dispersal. We refer to this third epoch of 1000 time steps as *Semi-Connected*. During the final 1000 time steps, the patches were each assigned one of two "habitat types" that conferred increased fitness to juveniles with specific genotypes. We refer to this landscape as *Semi-Connected with Local Selection*.

## Treatments and observable responses

Individual dispersal behavior (long vs. short), strength of selection (strong vs. weak), and barrier gap permeability (low vs. high) together formed eight treatment combinations. Each treatment was repeated in ten replicates. For each treatment, we tracked individual genotypes, per-capita homozygosity, per-patch population size, and the number of dispersers moving between the patches. We did not track individual pedigree information. Upon completion of the simulations, we used HexSim's report generator to create files suitable for input to the genetic software package STRUCTURE [36], which we subsequently used for some of our analyses.

Adult females in our model could reproduce each time step until their death, thereby producing overlapping generations. We measured generation time as the observed average age of reproducing females [37]. Migration rates between patches were measured during the *Semi-Connected* epoch. To qualify as a migrant, individuals had to cross through a barrier gap and subsequently reproduce somewhere other than their natal patch. Our migration data thus intentionally excluded non-breeders and individuals that reproduced in their natal patch after making a temporary excursion elsewhere.

## Landscape genetics

Isolation by distance (IBD) produced by dispersal limitations [38] is a core concept in landscape genetics. We used results from the *Continuous* epoch to illustrate the degree of IBD for our short and long-distance dispersal treatment groups. We constructed dispersal kernels by plotting the frequency of observed dispersal distances for all individuals in all replicates, from time step 1 to 1000. We visually assessed the degree of IBD using correlograms generated by the application *Alleles in Space* [39], where the average genetic distance between individuals is plotted for varying distance classes. We also plotted the frequency of observations for each distance class, to ensure that the inter-individual genetic distances were not due to underlying distribution of individuals on the landscape.

We calculated inter-individual genetic-distances resulting cumulatively from each landscape history. For ease of illustration, we subset our results as follows: for each treatment group, we randomly selected a single replicate simulation. From the selected replicate, we extracted genotype reports from the last time step of each of the four landscape epochs. From those reports, we randomly selected 25 individuals from each of the six patch locations. For those 150 individuals, we calculated a simple metric of pairwise genetic distance (number loci for which alleles differ between individuals / total number loci) using the "*ape*" R (v3.2.2) package "*dist.gene*" function with the "*percentage*" method [40, 41]. We visualized the 150 x 150 pairwise genetic distances as triangular matrices.

## Population genetics

We used a Bayesian clustering method, implemented by the program STRUCTURE (v 2.3.4), to analyze population genetic structure [36]. At the end of each landscape epoch, we collected genotype information stratified by putative population (patch), and imported these data into STRUCTURE. Unequal sample sizes (small patches ≈ 25 individuals, large patches ≈ 1000) interfered with the assessment of the number of subpopulations and assignment probabilities. This known limitation of the software [42] was easily overcome by randomly drawing a fixed number of individuals (n = 25) from each patch for analysis. In cases where there were fewer than 25 individuals extant in a given patch, genotypes were randomly selected for inclusion more than once until each patch had a sample of 25, for 150 total individuals.

We expected the number of unique genetic clusters (commonly referred to as "K") to vary from 1 to 6 based on landscape structure. Therefore, following convention, we tested possible values of K ranging from 1 to 20. All STRUCTURE analyses were run with a burn-in period of 10,000 iterations, with an additional 10,000 analysis iterations. We performed 20 replicate trials for each possible K value using the default settings of the admixture model and correlated allele frequencies. The best supported *K* values were identified using two methods: 1) plotting the replicate average *Ln P(D|K)*, and visually determining the minimum *K* of the curve's asymptote [42], and 2) using Evanno's Δ*K* method [43]. When there were close ties between supported *K* values from the competing methods, we considered them all for individual assignment analyses.

## Conservation biology

We calculated *Per-capita Homozygosity* as the percent of homozygous genotypes across all alleles and individuals within a given landscape patch. Large values of *Per-capita Homozygosity* could result from a few highly inbred individuals or from many slightly inbred individuals. We also examined the effect of patch isolation on allele frequencies from data collected at the ends of the *Continuous* and *Isolated Patches* epochs. We calculated three metrics of genetic degradation for each locus in each subpopulation using output from the "'*diveRsity*' R package "*divBasic*" function [44], as described below:

1. *Allelic richness* = the number of unique alleles per locus. All ten loci began with 5 alleles each.

2. *Allelic evenness* = $\frac{\sum p_i \ln(p_i)}{\ln(5)}$, where $p_i$ is the frequency of the $i^{th}$ allele within each locus, and *ln* (5) is the maximum evenness in each locus, given that all were initiated with 5 alleles. The starting population had different initial allele frequencies corresponding to allelic evenness values of 1.0 = "Equal", 0.96 = "Unequal", and 0.72 = "Rare" (Table 2).

3. *Heterozygosity deficit* = $H_{obs}$–$H_{exp}$, where $H_{obs}$ is the number of observed heterozygous genotypes and expected heterozygosity is calculated as: $H_{exp} = 1 - \sum_1^n (q_i)^2$, where *n* represents the number of alleles, and $q_i$ is the frequency of the $i^{th}$ allele at a locus. Note that $H_{exp}$ is calculated based on the number of extant alleles at a locus, and will fluctuate dramatically as alleles are lost from a population. Therefore, unlike *Allelic richness* or *Allelic evenness*, this metric has a shifting baseline.

For each metric, we calculated the mean and standard deviation across all replicates within the same patch size (small, medium, or large) and of the same initial allele frequency (equal, unequal, or rare) for each of the two time points of interest (at the end of the *Continuous* and *Isolated Patches* epochs).

### Evolutionary ecology

Population size was tracked for each patch across all time steps to facilitate measuring the response to adaptation. Additionally, we computed the change in frequency of adaptive alleles within each patch during the *Semi-Connected with Local Selection* epoch. L6 contained two adaptive alleles and 3 neutral, while L7-through L10 each contained one adaptive allele and 4 neutral. L6-A2, L7-A5 and L9-A1 were adaptive in half of the patches, while L6-A4, L8-A5 and L10-A1 were adaptive in the other half. Tracking allele frequencies in L6 allowed us to examine the effect of local adaptation on allele frequency in combination with asymmetric migration from adjacent patches, where the opposite allele was advantageous.

## Results

The age-class structure that arose from our model resulted in an average age of reproducing females of 8.7 years. Therefore, each landscape epoch of 1000 simulation time steps (Fig 1) spanned approximately 115 generations. During the *Semi-Connected* epoch, migration rate between adjacent patches varied with dispersal distance, barrier gap permeability, and patch size (Fig 2); but regardless of these values, the majority of individuals remained in their natal patch. As anticipated, we observed higher migration rates when individuals were assigned the long-distance dispersal behavior and the barrier gaps were highly permeable. Unequal perimeter-area ratios ensured that migration was asymmetric, with smaller patches generating proportionately greater numbers of emigrants than larger patches, and larger patches receiving proportionately higher numbers of immigrants than smaller patches.

### Landscape genetics

The short and long-distance dispersal behaviors measured during the *Continuous* landscape epoch produced notably different dispersal kernels (Fig 3 *left*). IBD was evident in the relationship between geographic distance and genetic distance when dispersal was short, but not when dispersal distances were long (Fig 3 *right*).

We also interpret our series of landscape epochs as four nested hypothetical landscape histories: years 1–1000, years 1–2000, years 1–3000, and years 1–4000. Genetic distance matrices

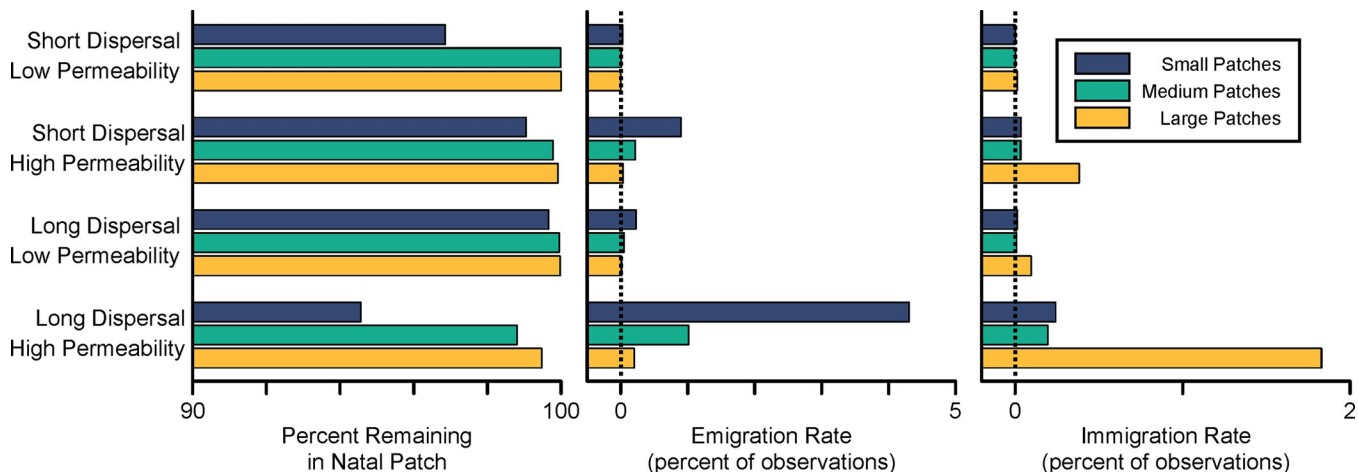

**Fig 2. Observed migration rates of long or short-dispersing individuals across barriers of varying permeability during the *Semi-Connected* epoch.** Left: Percent of the population that is born and reproduces in the same patch. Middle: Emigration rates, sorted by size of sending patch. Right: Immigration rates, sorted by size of receiving patch.

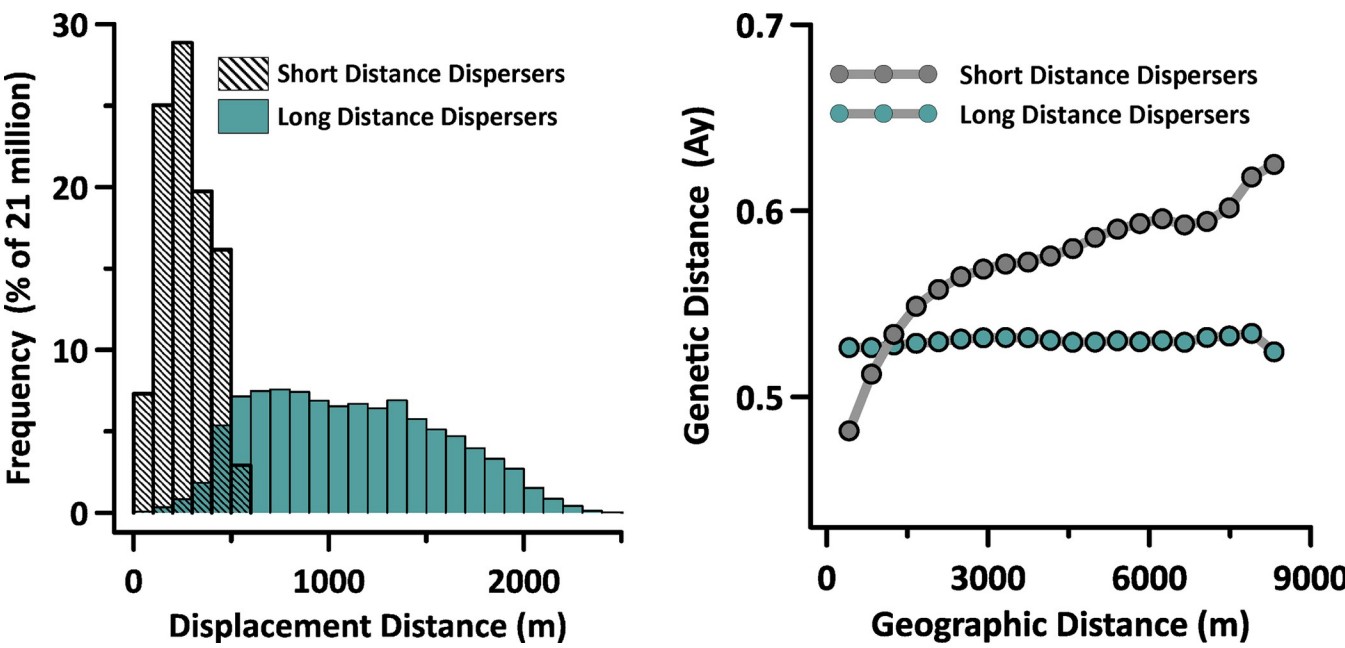

**Fig 3.** In a spatially-continuous landscape, the observed dispersal kernels reflected juvenile dispersal ability (left). Genetic IBD was evident when dispersal paths were short, but not when they became longer (right). These data were gathered at the end of the *Continuous* landscape epoch, after 115 generations of unobstructed movement.

observed at the conclusion of each epoch illustrate the effectiveness of gene-flow at mixing subpopulations, given landscape history and dispersal behavior (Fig 4). Additionally, barrier gap permeability affects gene-flow in the *Semi-Connected* and *Semi-Connected with Local Selection* epochs, and selection strength affects gene-flow in the *Semi-Connected with Local Selection epoch.*

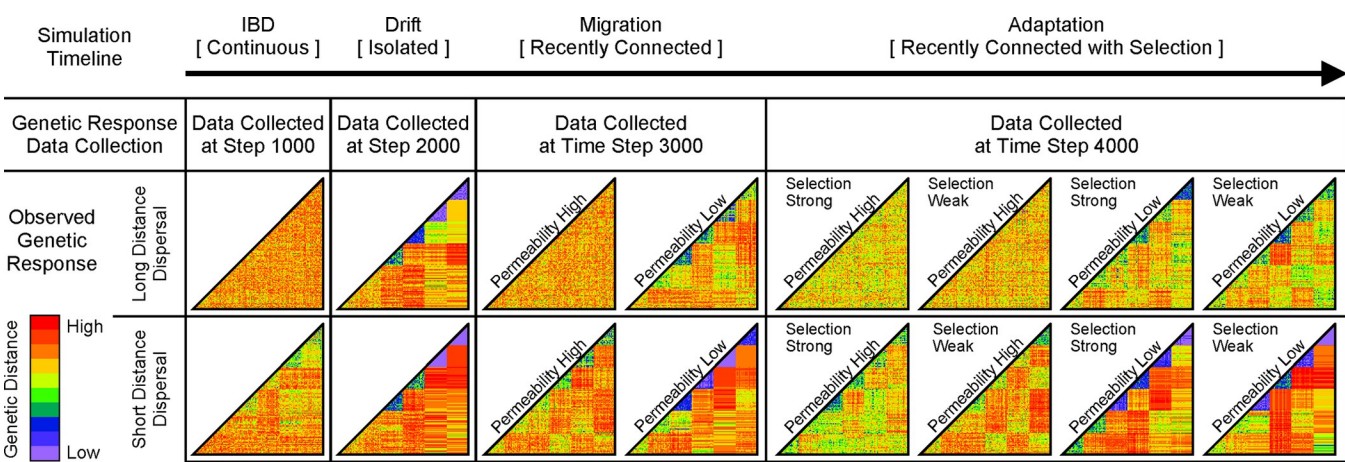

**Fig 4. Modeling gene-flow for four landscape histories, beginning with IBD only, and building to a complex history of sequential epochs incorporating IBD, drift, migration, and finally, local selection.** The resulting pairwise genetic distance matrices (colored triangles) illustrate the strength of gene-flow within a given landscape history, where increasing contrast in the geometric patterns visible within the matrices results from reduced gene-flow between landscape patches. Data collected at time step 4000 illustrate the interacting effects of gene-flow and local adaptation on genetic-distance. See Methods section for additional details.

**Table 3. Inferred number of genetic subpopulations *K* resulting from either the visual interpretation of likelihood probabilities *L(K)* or from Evanno's ΔK method *E*, as a function of dispersal distance and epoch.** When a single method produced multiple equally-supported estimates of K, then all values were included.

|  | K | Continuous Epoch | Isolated Patches Epoch |
|---|---|---|---|
| **Long Distance Dispersal** | 1 | L(K) | |
|  | 2 | E | |
|  | 5 | | L(K) |
|  | 6 | | L(K) |
|  | 8 | | E |
| **Short Distance Dispersal** | 2 | E | |
|  | 4 | L(K) | |
|  | 5 | L(K) | L(K) |
|  | 6 | | L(K) |
|  | 7 | | E |

## Population genetics

In the *Continuous* epoch, we know a single biological population was present on the landscape. But without the benefit, inherent in modeling, of knowing the true population dynamics, genetic analysis could easily lead to a conclusion that several distinct populations were present (Table 3). For example, neither Evanno's method or L(*K*) produced consistent results in this case. Short distance dispersal and the resulting IBD led to an inference of multiple genetic clusters under L(*K*), but a single genetic cluster under *E*. When dispersers moved longer distances, and IBD was therefore absent, the L(*K*) method produced the expected result, but Evanno's method did not (Table 3).

In the *Isolated Patches* epoch, there are 6 distinct populations isolated by absolute barriers. By the end of the epoch, 5–8 unique genetic clusters were identified by our genetic analyses, regardless of dispersal behavior (Table 3). Evanno's method (*E*) tended to produce inflated estimates of *K* that exceeded the true number of 6 genetic subpopulations. The STRUCTURE plots (not shown) indicated clearly that the 4 smaller patches were each home to a single genetic cluster, made unique by drift. The two large patches housed either one or two genetic clusters, due to a weak effect of drift in these larger populations.

The population genetic structure produced by drift during the *Isolated Patches* epoch was mitigated by the limited inter-patch migration that characterized the subsequent *Semi-Connected* epoch (Table 4). The differing numbers of migrants per generation produced varying degrees of genetic mixing over the course of these 115 generations. The observed >15 migrants per generation resulting from long distance dispersal and high barrier gap permeability produced sufficient genetic mixing to return to panmixia, while the other treatments failed to do so. Long-distance dispersal with low barrier permeability resulted in <1 migrant per

**Table 4. The replicated average number of observed migrants per generation, presented as a function of dispersal distance and barrier gap permeability, at the end of the *Semi-Connected* epoch.** In most cases, population genetic structure (change in possible *K*, evaluated from either the visual interpretation of likelihood probabilities *L(K)* (top) or from Evanno's ΔK method *E* (bottom)) declined from where it began in the prior *Isolated Patches* epoch.

|  | Low Barrier Gap Permeability | | High Barrier Gap Permeability | |
|---|---|---|---|---|
|  | Migrants per Generation | Change in Possible K | Migrants per Generation | Change in Possible K |
| **Long Distance Dispersal** | 0.757 | 5 or 6 → 4 | 15.456 | 5 or 6 → 1 |
|  |  | 8 → 6 |  | 8 → 2 |
| **Short Distance Dispersal** | 0.086 | 5 or 6 → 6 | 2.598 | 5 or 6 → 6 |
|  |  | 7 → 2 |  | 7 → 2 |

generation, and this migration rate was insufficient to return the population to panmixia within the 115 generations of the epoch. A return to panmixia by the end of the *Semi-Connected* epoch was not expected in simulations employing a short dispersal distance, and no such outcome was observed.

## Conservation biology

In this analysis, which included only the *Continuous*, *Isolated Patches*, and *Semi-Connected* epochs, per-capita homozygosity varied dramatically with landscape epoch, and fluctuated more in the small patches than the large ones (Fig 5). While we observed little accumulation of homozygosity during the *Continuous* epoch, this changed when migration between patches was limited by dispersal barriers. During the *Isolated Patches* epoch, homozygosity increased rapidly, but this general trend was strongly influenced by patch size. Both patch size and barrier gap permeability affected the amount of "genetic rescue" resulting from migration during the *Semi-Connected* epoch.

Changes in allele frequencies observed during the *Continuous* and *Isolated Patches* epochs were affected by patch size, locus-specific initial allele frequencies, and dispersal distance. Even during the *Continuous* epoch, prior to patch isolation, rare alleles were lost due to drift. As anticipated, this process became more pronounced when dispersal was limited by impermeable barriers (the *Isolated Patches* epoch). The loss of allelic richness and evenness during patch isolation was most pronounced in the small patches, and least severe in the large patches. Short-distance dispersal also influenced loss of allelic richness and evenness in both epochs, but only in medium and small patches. A heterozygosity deficit was observed when dispersal distances were short, but only in large and medium patches.

## Evolutionary ecology

Selection pressure was initiated at the beginning of the final landscape epoch, at time step 3000. Effective carrying capacity and population sizes emerged mechanistically from our model. Population size was most stochastic in the smaller patches, and most stable in the large patches (Fig 6). A nonzero probability of extinction was observed only for the small patches, and was greatest when the dispersal distances were short and movement barriers were present. The simulated adaptation to local conditions (time steps 3000–5000) effectively increased patch carrying capacity. These effects of adaptation were more pronounced when selection pressure was strong.

The frequency of the adaptive alleles at loci L7 and L10 increased globally across all subpopulations as selection acted locally and migration moved the adaptive alleles across the landscape. However, the effect of local selection at Locus L6 was counteracted by asymmetric migration (see Fig 2) between adjacent patches of different habitat types. In small patches, local selection was almost always swamped by migration from the neighboring large patches, except when selection was strong, dispersal distance was short, and gap permeability was low. Otherwise, for the small and medium patches, the extent to which selection was swamped by migration varied more continuously based on the strength of selection, barrier gap permeability, and dispersal distance. In large patches, selection was often able to act effectively regardless of selection strength or migration rate, except when selection was weak and dispersal distance was long.

## Discussion

We attempted to develop the simplest individual-based model capable of clearly illustrating how spatial structure drives eco-evo dynamics. By imposing landscape change on otherwise

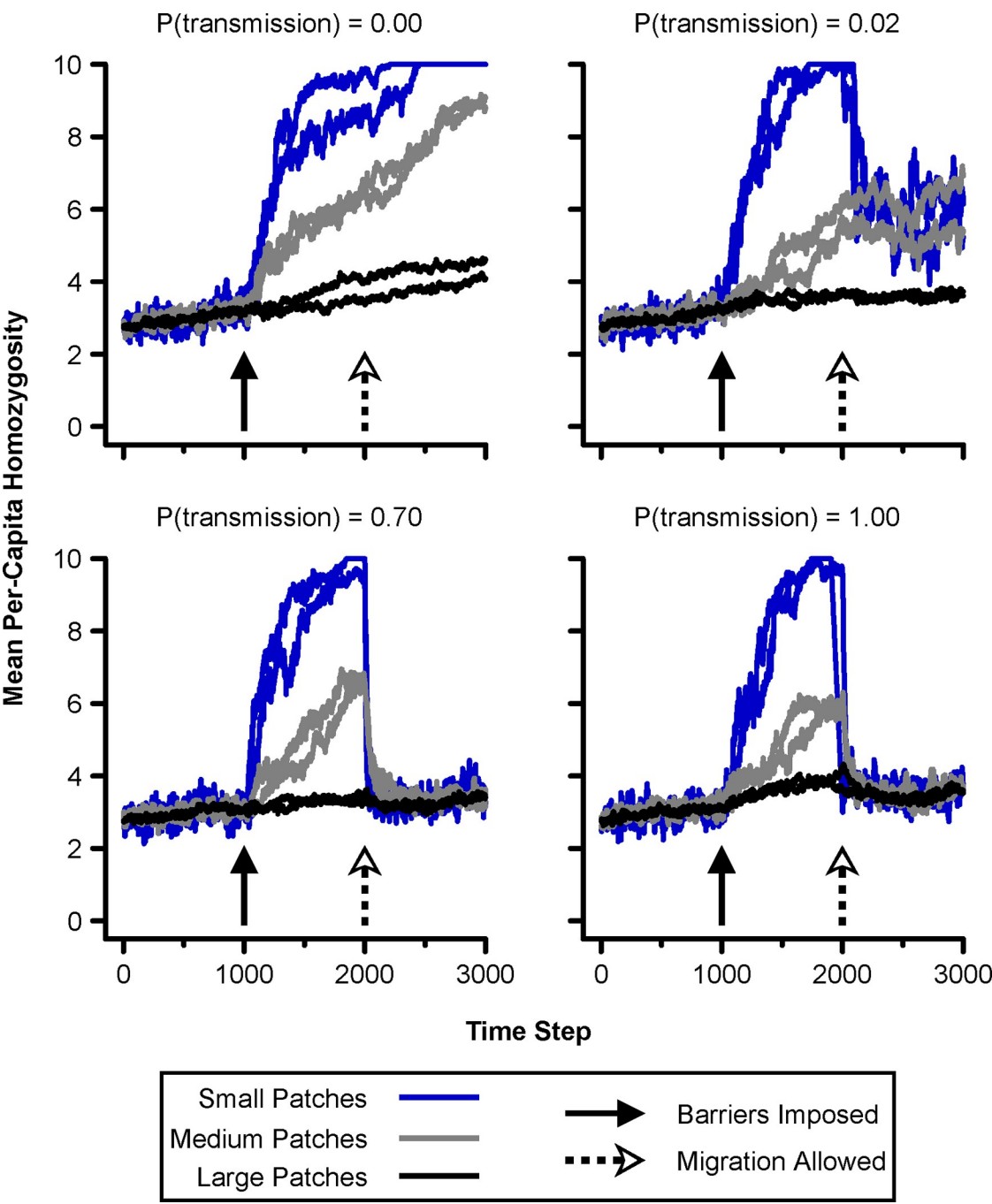

**Fig 5. Per-capita homozygosity across simulation time steps spanning the *Continuous, Isolated Patches*, and *Semi-Connected* landscape epochs, displayed by patch size.** Our simulated low and high barrier crossing probabilities corresponded to P (transmission) values of 0.02 and 0.70, respectively. Six lines per plot result from two small, two medium, and two large patches.

functionally-static eco-evolutionary models, we were able to alter key emergent forces such as gene-flow, genetic drift, and adaptive selection, as well as demo-genetic responses including population size, probability of extinction, and allele frequencies. Our model also demonstrated how critical demo-genetic traits, for example generation time and migration rate, can arise from a parsimonious mechanistic model, rather than being specified *a priori*.

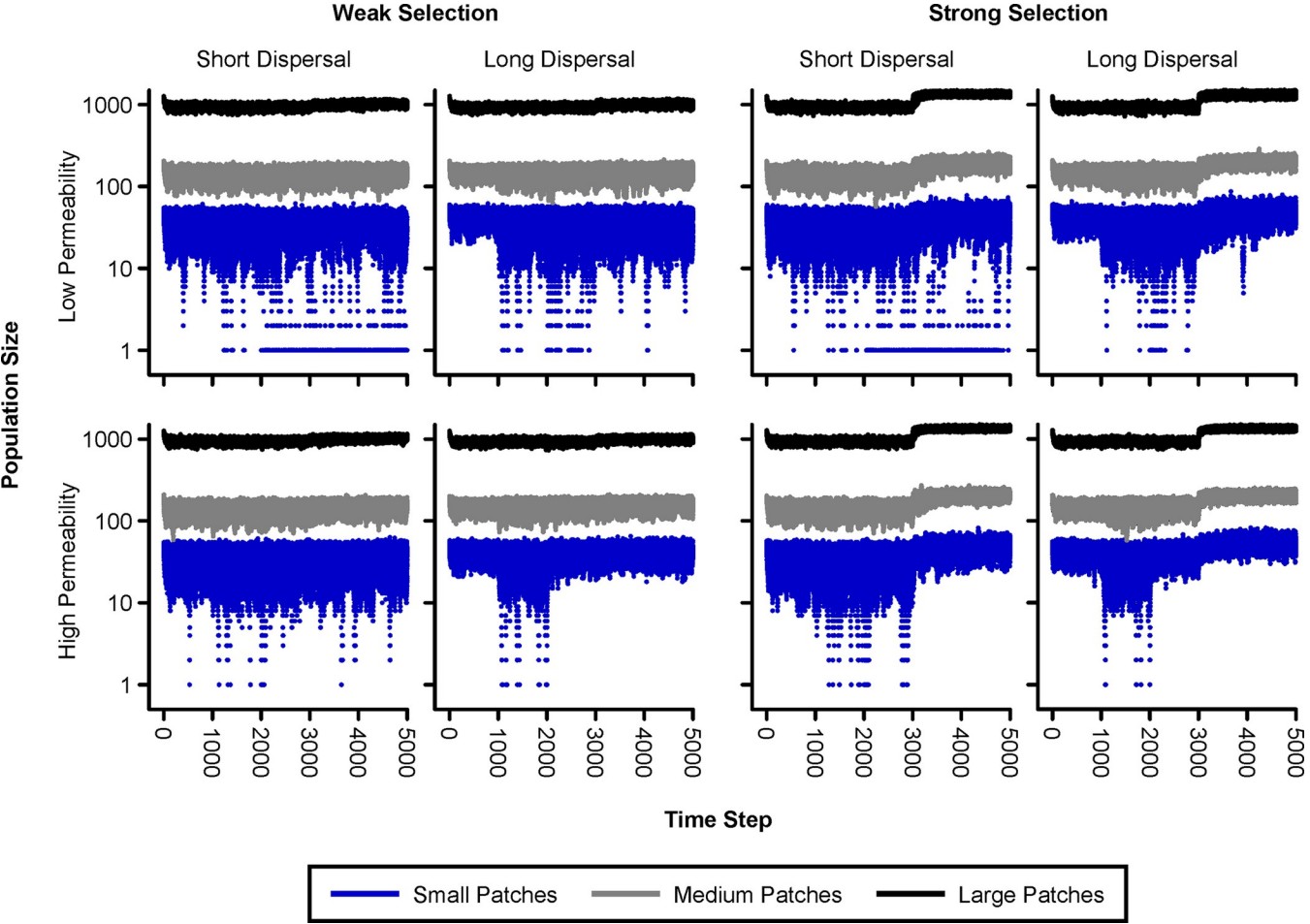

**Fig 6. Observed trends in population size stratified by patch size.** Results from all ten simulation replicates are shown. Data are displayed for an additional 1000 time steps during the final landscape epoch in order to better visualize the long-term population response to selection pressure. Carrying capacity was effectively increased by local adaptation, but the magnitude of this effect depended on the strength of selection. Extinction events can be inferred from occasional very low population sizes observed for the small patches (note the logarithmic vertical axis).

## Landscape genetics

A common approach in landscape genetics is to assume that the current genetic pattern is a result of landscape pattern (with some lag-time) represented by resistance surfaces. Hypothetical "resistance distances" between individuals are compared to actual genetic distances between sampled individuals (or groups), and tests are performed to infer which elements in the landscape most strongly influence gene-flow. But complex movement behavior and dynamic landscape histories are not easily captured by resistance surfaces, and this in turn limits our ability to identify causal processes and detect past gene-flow [45, 46].

We demonstrated how the cumulative effects of gene-flow, genetic drift, and selection can be simulated over several hypothetical landscape histories. The resulting inter-individual genetic distance matrices simultaneously captured the influences of dynamic landscape structure, dispersal behavior, and demography. Long-distance dispersal behavior tended to swamp any effect of landscape history when landscape connectivity was high. This suggests it may be difficult to identify past landscape discontinuities when dispersal distances are long relative to the spatial extent of the study area.

## Population genetics

An ongoing challenge for empirical population geneticists and phylogeneticists has been determining the number of populations via analysis of genetic clustering patterns. Our model demonstrates that small amounts of IBD inflate such inferences, and that popular statistical solutions for choosing between possible numbers of genetic clusters are prone to overestimation [47].

Our approach offers population geneticists a methodology that allows migration rates to emerge from species-landscape interactions. Our model demonstrates how a simple alteration of barrier gap permeability can produce departures from the "one-migrant-per-generation" theoretically-derived guideline for re-establishing genetic panmixia [48]. We observed that, with short-distance dispersal behavior, the inference of population structure was similar when that structure resulted from either IBD or from gene-flow following a period of isolation, highlighting the difficulty in disentangling past genetic history from contemporary genetic structure.

## Conservation biology

Forecasting genetic degradation in small populations facing extinction is a challenging task that should ideally acknowledge the influences of both demographic and genetic processes. Our methodology addresses this challenge, as illustrated by two examples from our results. First, rare alleles were lost from the small patches, even when the landscape was fully connected; and the severity of this loss was more extreme when individuals dispersed short distances. Second, the loss of allelic evenness we observed exhibited different patterns depending on patch size. In the small patches, the allelic evenness eventually approached zero, due to single allele fixation, regardless of initial allele frequencies or dispersal distance. Conversely, in the large patches, allelic evenness varied depending upon initial conditions, but did not vary with dispersal distance. We observed a combined effect of dispersal distance and initial conditions in the medium patches, where limiting dispersal distance increased the loss of allelic evenness, but the steady-state depended on initial conditions. Traditional simulation approaches would associate a single trend with any given landscape.

These findings highlight how eco-evo simulators can add realism to the forecasts of genetic degradation used in conservation, and demonstrate how future studies might improve our understanding of inbreeding and outbreeding depression, better explore the relative importance of genetic versus demographic components of viability, and more.

## Evolutionary ecology

The HexSim modeling platform allows users to develop dynamic feedback loops linking evolutionary and ecological forces. Our model captured the counteracting forces of local selection and asymmetric migration from spatially proximal sub-populations. Adaptive alleles increased in frequency in spatially discontinuous landscape patches due to local selection, while migration rates between patches varied by patch size, dispersal behavior, and barrier gap permeability. The increase in adaptive alleles was most pronounced when selection was strong, yet this effect was not observed in the small and medium-sized patches when dispersal distance was short and barrier gap permeability was low, presumably because stochastic drift was then the driving force. When selection was weak, asymmetric migration between the large and proximate smaller patches produced highly variable changes in the frequency of the locally adaptive allele in the smaller patches. Even in the large patches, where in the absence of drift, selection might be expected to act efficiently, the anticipated changes in adaptive alleles were only observed when dispersal ability was limited and barrier gap permeability was low. The

interplay between local selection and asymmetric migration that we found in our smaller patches demonstrates what we might expect to see as species' ranges shift due to climate change.

## Conclusions

Contemporary ecological theory reflects a scientific world-view heavily influenced by mathematics and simulation models. But these useful abstractions only have practical value when equations can be formulated and solved, or source code can be designed and executed; and as is true with science in general, ecologists have made use of simplifying assumptions in order to keep model development tractable. The cost of these simplifications has been that our theory can lack rigorous grounding in the very biological detail that we know governs species' interactions with their environments, and with each other. Simplifying assumptions are a practical necessity, but we should resist becoming too comfortable with them.

Our study used a two-pronged approach to examine these issues. First, we illustrated how advanced software tools (HexSim, in our case) allow researchers to explore complex biological mechanisms while retaining species-landscape and intraspecific interactions. Second, we examined specific simplifying assumptions that have shaped the development of theories in landscape genetics, population genetics, conservation biology, and evolutionary ecology. For each discipline, we used our hypothetical model system to illustrate how renewed attention to pattern and process can highlight the limits of existing theory. Our objective is not to find fault in past work, but rather to improve our models by challenging the assumptions upon which they have been constructed. To be sure, eco-evo theories of the future will extend existing models, not discard them. And the next generation of eco-evo theory will better explain patterns and processes when landscapes are spatially and temporally dynamic, species' life histories and genetics are complex and interacting, and when the details matter.

The simulations described here are, by design, simple and straightforward. But they illustrate fundamental biological forces, track familiar observable responses, and address pressing challenges in the fields of landscape genetics, population genetics, conservation biology, and evolutionary ecology. We also expect our methods to have applications in the fields of phylogenetics, phylogeography, medical science, and evolutionary theory. For example, phylogeneticists might simulate past species' range shifts, model incomplete lineage sorting, or ask how linkage affects coalescent metrics. Phylogeographers may explore the stability of admixture zones over time, or improve the demarcation of evolutionary significant units. More generally, future applications of our methods will benefit from the HexSim modeling platform's ability to incorporate dynamic geographically realistic landscapes, and to simulate real species, interactions, and disturbance regimes.

While our simulations included only simple microsatellite-like genotypes, we anticipate future projects will investigate how mode of inheritance, linkage between alleles, initial geographic distributions of allele frequencies, and multiple mutation regimes (e.g., stepwise, infinite alleles, or two-phase models) interact with landscape spatial pattern and movement. Conservation geneticists may simulate mutation rates based on exposure to spatially-distributed chemical mutagens encountered during dispersal and mitigated by selection against deleterious alleles. Similarly, rates and patterns of gene-flow may be investigated within the context of nuanced, biologically-realistic movement behaviors, including rare long-distance dispersal events, site fidelity and memory, attraction and avoidance, sex-specific behaviors, to name a few. HexSim's design makes all of these simulation strategies possible, and more, while still being an ideal platform for simple models like the one explored in this study. Our methodology can also be extended to capture complex intermediate relationships between genotype,

phenotype, and fitness, thus opening the door to evolutionary modeling at the level of genes, phenotypic heritable traits, populations, or even communities.

## Acknowledgments

Allen Brookes developed the source code for HexSim and has been an invaluable team member. David Beck at the University of Washington eScience Institute aided in data management and processing. The information in this document has been subjected to review by the US EPA Center for Public Health and Environmental Assessment, and approved for publication.

## Author Contributions

**Conceptualization:** Jennifer M. White, Nathan H. Schumaker.

**Formal analysis:** Nathan H. Schumaker, Rachel Y. Chock, Sydney M. Watkins.

**Investigation:** Jennifer M. White, Nathan H. Schumaker.

**Methodology:** Jennifer M. White, Nathan H. Schumaker.

**Project administration:** Jennifer M. White, Nathan H. Schumaker.

**Software:** Nathan H. Schumaker.

**Validation:** Rachel Y. Chock, Sydney M. Watkins.

**Visualization:** Rachel Y. Chock, Sydney M. Watkins.

**Writing – original draft:** Jennifer M. White, Nathan H. Schumaker.

**Writing – review & editing:** Jennifer M. White, Nathan H. Schumaker, Rachel Y. Chock, Sydney M. Watkins.

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
