## [Decision Letter · Decision Letter 0]

21 Nov 2022

PONE-D-22-20501Adding Pattern and Process to Eco-Evo Theory and Applications.PLOS ONE

Dear Dr. Schumaker,

Thank you for submitting your manuscript to PLOS ONE. After careful consideration, we feel that it has merit but does not fully meet PLOS ONE’s publication criteria as it currently stands. Therefore, we invite you to submit a revised version of the manuscript that addresses the points raised during the review process.

We look forward to receiving your revised manuscript.

Kind regards,

Pankaj Bhardwaj, Ph.D.

Academic Editor

PLOS ONE

Journal Requirements:

2. Please remove your figures from within your manuscript file, leaving only the individual TIFF/EPS image files, uploaded separately. These will be automatically included in the reviewers’ PDF.

Reviewer's Responses to Questions

**Comments to the Author**

1. Is the manuscript technically sound, and do the data support the conclusions?

Reviewer #1: Partly

Reviewer #2: Yes

2. Has the statistical analysis been performed appropriately and rigorously? 

Reviewer #1: I Don't Know

Reviewer #2: Yes

3. Have the authors made all data underlying the findings in their manuscript fully available?

Reviewer #1: Yes

Reviewer #2: Yes

4. Is the manuscript presented in an intelligible fashion and written in standard English?

Reviewer #1: Yes

Reviewer #2: Yes

5. Review Comments to the Author

Reviewer #1: 1. The research paper entitled “Adding Pattern and Process to Eco-Evo Theory and Applications. Short Title: Eco-Evo Theory and Applications” by Schumaker et al. presented a novel simulation modeling approach for investigating eco-evolutionary dynamics, centered on the driving role of landscape pattern. The manuscript is hard to catch what is the exact subject of the study described.

2. The introduction is also vague and somewhat confusing, and is not organized in a scientific fashion. What is the background of the research, why does it need to be investigated?

3. Introduction and conclusion are not at par with the hypothesis.

4. The methods are not clear and are very difficult to understand. It could be a little simple and illustrative.

5. Discussion can be improved.There is no crosstalk in the discussion. You can discuss with some of the cross-references then it will make more sense of the study.

6. The overall text is not written clearly. There is too much of wordiness in the lines. It should be removed to make it simple and crisp.

7. The HexSim approach that you used, has a global significance in conservation management for in particular or for any ecosystem in general?

Reviewer #2: Eco-evolutionary and the lesser understood, landscape dynamics have a complex relationship that influence population responses. The authors have attempted to provide an improved simulation modeling approach to understand these phenomena. They further tested and altered fundamental assumptions and patterns of evolutionary forces and thereafter, observed responses to these manipulations. In my opinion, this manuscript and the resulting simulation has the potential to provide further insights in the existing eco-evolutionary theory and henceforth, its applications.

6. PLOS authors have the option to publish the peer review history of their article (what does this mean?). If published, this will include your full peer review and any attached files.

Reviewer #1: **Yes: **Amandeep Singh

Reviewer #2: No

---

## [Author Response · Author response to Decision Letter 0]

1 Dec 2022

Please see the attached PDF file titled Response to Reviewers.

---

## [Decision Letter · Decision Letter 1]

2 Jan 2023

PONE-D-22-20501R1Adding Pattern and Process to Eco-Evo Theory and ApplicationsPLOS ONE

Dear Dr. Schumaker,

Thank you for submitting your manuscript to PLOS ONE. After careful consideration, we feel that it has merit but does not fully meet PLOS ONE’s publication criteria as it currently stands. Therefore, we invite you to submit a revised version of the manuscript that addresses the points raised during the review process.

There are few minor issues highlighted by the reviewer 1. please go through the attached copy of the comments and address in detail. Please submit your revised manuscript by Feb 16 2023 11:59PM. If you will need more time than this to complete your revisions, please reply to this message or contact the journal office at plosone@plos.org. Please include the following items when submitting your revised manuscript:A rebuttal letter that responds to each point raised by the academic editor and reviewer(s). You should upload this letter as a separate file labeled 'Response to Reviewers'.A marked-up copy of your manuscript that highlights changes made to the original version. You should upload this as a separate file labeled 'Revised Manuscript with Track Changes'.An unmarked version of your revised paper without tracked changes. You should upload this as a separate file labeled 'Manuscript'.If applicable, we recommend that you deposit your laboratory protocols in protocols.io to enhance the reproducibility of your results. Protocols.io assigns your protocol its own identifier (DOI) so that it can be cited independently in the future. For instructions see: https://journals.plos.org/plosone/s/submission-guidelines#loc-laboratory-protocols. Additionally, PLOS ONE offers an option for publishing peer-reviewed Lab Protocol articles, which describe protocols hosted on protocols.io. Read more information on sharing protocols at https://plos.org/protocols?utm_medium=editorial-email&utm_source=authorletters&utm_campaign=protocols.

We look forward to receiving your revised manuscript.

Kind regards,

Pankaj Bhardwaj, Ph.D.

Academic Editor

PLOS ONE

Journal Requirements:

Reviewers' comments:

Reviewer's Responses to Questions

**Comments to the Author**

1. If the authors have adequately addressed your comments raised in a previous round of review and you feel that this manuscript is now acceptable for publication, you may indicate that here to bypass the “Comments to the Author” section, enter your conflict of interest statement in the “Confidential to Editor” section, and submit your "Accept" recommendation.

Reviewer #1: All comments have been addressed

Reviewer #2: All comments have been addressed

2. Is the manuscript technically sound, and do the data support the conclusions?

Reviewer #1: Yes

Reviewer #2: Yes

3. Has the statistical analysis been performed appropriately and rigorously? 

Reviewer #1: Yes

Reviewer #2: Yes

4. Have the authors made all data underlying the findings in their manuscript fully available?

Reviewer #1: Yes

Reviewer #2: Yes

5. Is the manuscript presented in an intelligible fashion and written in standard English?

Reviewer #1: Yes

Reviewer #2: Yes

6. Review Comments to the Author

Reviewer #1: (No Response)

Reviewer #2: The authors have addressed the comments raised by the reviewers and I recommend the article for publication.

7. PLOS authors have the option to publish the peer review history of their article (what does this mean?). If published, this will include your full peer review and any attached files.

Reviewer #1: No

Reviewer #2: No

---

## [Author Response · Author response to Decision Letter 1]

4 Jan 2023

Please see attached file "Response to Reviewers".

---

## [Decision Letter · Decision Letter 2]

3 Feb 2023

PONE-D-22-20501R2Adding Pattern and Process to Eco-Evo Theory and ApplicationsPLOS ONE

Dear Dr. Schumaker,

Thank you for submitting your manuscript to PLOS ONE. As you have opposed one of the reviewers from the earlier list, I have to restore a new reviewer who has suggested some additional changes. Please prepare a comment wise list of your response and resubmit the manuscript.

We look forward to receiving your revised manuscript.

Kind regards,

Pankaj Bhardwaj, Ph.D.

Academic Editor

PLOS ONE

Journal Requirements:

Reviewers' comments:

Reviewer's Responses to Questions

**Comments to the Author**

1. If the authors have adequately addressed your comments raised in a previous round of review and you feel that this manuscript is now acceptable for publication, you may indicate that here to bypass the “Comments to the Author” section, enter your conflict of interest statement in the “Confidential to Editor” section, and submit your "Accept" recommendation.

Reviewer #3: All comments have been addressed

2. Is the manuscript technically sound, and do the data support the conclusions?

Reviewer #3: Yes

3. Has the statistical analysis been performed appropriately and rigorously? 

Reviewer #3: Yes

4. Have the authors made all data underlying the findings in their manuscript fully available?

Reviewer #3: Yes

5. Is the manuscript presented in an intelligible fashion and written in standard English?

Reviewer #3: Yes

6. Review Comments to the Author

Reviewer #3: The authors present the lastest development of an agent-based model to simulate spatially explicit eco-evo processes and patterns. The manuscript is well-written, clear and easy to go through. One very big limitation I see (to be fair, the only major limitation) stands in the lack of context. Other tools out there simulate eco-evolutionary dynamics. Not all of them are agent-based, not all of them include genetic variability as part of the toolbox, but this does not mean HexSim could be thrown out without taking care of context. I'm not saying they should perform any statistical comparison with competing tools, but a minimal guidance offered to the readers about what the tool does in comparison to others is probably very useful, and I'd say a much needed addition to the present submission. I've offered some hints in the attached revision. The list there is minimal, but could serve the goal of pitting HexSim against competing approaches giving the necessary context.

regards

7. PLOS authors have the option to publish the peer review history of their article (what does this mean?). If published, this will include your full peer review and any attached files.

Reviewer #3: No

---

## [Author Response · Author response to Decision Letter 2]

13 Feb 2023

Please see the attached document "Response to Reviewers".

---

## [Decision Letter · Decision Letter 3]

17 Feb 2023

Adding Pattern and Process to Eco-Evo Theory and Applications

PONE-D-22-20501R3

Dear Dr. Schumaker,

We’re pleased to inform you that your manuscript has been judged scientifically suitable for publication and will be formally accepted for publication once it meets all outstanding technical requirements.

Kind regards,

Pankaj Bhardwaj, Ph.D.

Academic Editor

PLOS ONE

Additional Editor Comments (optional):

Reviewers' comments:

Reviewer's Responses to Questions

**Comments to the Author**

1. If the authors have adequately addressed your comments raised in a previous round of review and you feel that this manuscript is now acceptable for publication, you may indicate that here to bypass the “Comments to the Author” section, enter your conflict of interest statement in the “Confidential to Editor” section, and submit your "Accept" recommendation.

Reviewer #3: All comments have been addressed

2. Is the manuscript technically sound, and do the data support the conclusions?

Reviewer #3: Yes

3. Has the statistical analysis been performed appropriately and rigorously? 

Reviewer #3: Yes

4. Have the authors made all data underlying the findings in their manuscript fully available?

Reviewer #3: Yes

5. Is the manuscript presented in an intelligible fashion and written in standard English?

Reviewer #3: Yes

6. Review Comments to the Author

Reviewer #3: I'm happy with the review. With the (admittedly minimal) revision requested, the manuscript appears more balanced and fair with competing methods now.

I'm glad to recommend publication

regards

7. PLOS authors have the option to publish the peer review history of their article (what does this mean?). If published, this will include your full peer review and any attached files.

Reviewer #3: No

---

## [Editor Report · Acceptance letter]

27 Feb 2023

PONE-D-22-20501R3 

Adding pattern and process to eco-evo theory and applications 

Dear Dr. Schumaker:

I'm pleased to inform you that your manuscript has been deemed suitable for publication in PLOS ONE. Congratulations! Your manuscript is now with our production department. 

Kind regards, 

on behalf of

Dr. Pankaj Bhardwaj 

Academic Editor

PLOS ONE